# Assessing the Impact of Unfolding Case Study Scenarios during High-Fidelity Pediatric Simulation among Undergraduate Nursing Students

**DOI:** 10.3390/healthcare9111584

**Published:** 2021-11-19

**Authors:** Allison C. Munn, Beth Lay, Tiffany A. Phillips, Tracy P. George

**Affiliations:** Department of Nursing, Francis Marion University, Florence, SC 29506, USA; amunn@fmarion.edu (A.C.M.); elay@fmarion.edu (B.L.); tphillips@fmarion.edu (T.A.P.)

**Keywords:** simulation, pediatric nursing, pre-licensure nursing education, clinical education, nursing education research

## Abstract

Simulation helps to prepare prelicensure nursing students for practice by providing opportunities to perform clinical skills and make decisions in a safe environment. The integration of nursing knowledge, skills, and decision-making abilities during simulated unfolding case-study scenarios may enhance student self-confidence and foster clinical judgement skills. The purpose of this study was to assess the impact of simulation using unfolding case-study scenarios on undergraduate nursing students’ self-confidence in pediatric nursing knowledge, skills, and clinical judgment/decision-making abilities. This mixed methods study included a pre- and post-survey design to evaluate undergraduate nursing students’ confidence in pediatric nursing knowledge, skills, and decision-making abilities after participation in both an instructor-led (guided) and a student-led (decision-making) simulation involving unfolding case-study scenarios. Friedman’s ANOVA analyses revealed that all 16-items demonstrated statistically significant differences between the three measured responses (pre-simulation and both post-simulation surveys). Post-hoc Wilcoxon signed-rank tests revealed statistically significant differences in student ratings pre-simulation and post-instructor-led (guided) experience for all 16-scored items. The qualitative themes identified were perception of experience, pediatric nursing care, assimilation of knowledge, and critical thinking. Unfolding case-study simulation experiences positively impact the learning, self-confidence, and clinical judgement of undergraduate nursing students.

## 1. Introduction

Simulation is a teaching technique that allows for student immersion in a guided learning activity or environment using dramatization of real-world situations and scenarios [1,2]. Simulation in healthcare education provides an opportunity for students to make patient care decisions while functioning in a safe and controlled environment [2]. Nursing students often experience anxiety during clinical experiences [3,4]. Diversification of learning experiences through a simulated scenario can help students to improve clinical judgement skills while also providing an opportunity for instructors to therapeutically discuss and allay student fears and anxieties [3,4,5]. The World Health Organization (WHO) supports the use of simulation in healthcare training and education programs to improve students’ skills, self-confidence, communication, teamwork, and decision-making capabilities [2,6].

Entry-level nurses are required to care for higher acuity patients than in previous years. New graduates’ ability to organize information, think critically, and use sound clinical judgment in a variety of patient situations is an expectation [7]. The National Council State Boards of Nursing (NCSBN) is moving toward the Next Generation NCLEX which assesses clinical judgment [8]. These changes to the exam are supported by the Clinical Judgment Model (CJM), which includes the following components: recognizing and analyzing cues, prioritizing hypotheses, generating solutions, taking action, and evaluating outcomes. Clinical judgment assists students as they transition to the role of the registered nurse in practice [8]. According to the Carnegie Report, simulation is a recommended way to link the classroom and clinical practice, and it allows students to utilize the Clinical Judgment Model in patient care scenarios [8,9]. Through high-fidelity simulation, faculty are able to assess students’ clinical judgment and provide constructive, real-time feedback.

Simulation in nursing education can take on many different forms and is paramount to the production of well-prepared registered nurses who can incorporate skill and critical thinking into practice during emergent and critical patient care situations [10,11,12]. Additionally, simulation has been found to be an effective teaching strategy, allowing faculty and students to assess nursing knowledge, critical thinking, and technical skills [13]. Incorporating simulation into nursing curriculums has been positively received by students and has been shown to improve student knowledge and self-confidence and to contribute to reflection and meaningful learning. Attainment of knowledge and skill through simulation improves student performance, lessens anxiety about critical patient situations, improves the learner’s ability to function in critical situations, and contributes to patient safety [10,12].

Pediatric simulation allows students to apply knowledge obtained in the didactic environment to a clinical scenario and to obtain faculty feedback [13,14]. Lubbers and Rossman reported that undergraduate nursing students who participated in pediatric simulations had increased self-confidence and reported high satisfaction after their simulation experiences [15,16]. Similarly, Parker and colleagues found that undergraduate students in a pediatric simulation gained more confidence in clinical skills and were pleased with their participation in medium- and high-fidelity clinical simulations [14]. Furthermore, pediatric simulation can allow for exposure to clinical scenarios and situations that nursing students may not have the opportunity to experience in the traditional clinical setting.

During the COVID-19 pandemic, placements in pediatrics clinics have become more difficult. Thus, simulation has emerged as an important learning modality that allows students to gain the knowledge and skills necessary to function after graduation in the role of a registered nurse in pediatric settings.

The purpose of this study was to assess the impact of simulation using unfolding case-study scenarios on undergraduate nursing students’ self-confidence in pediatric nursing knowledge, skills, and clinical judgment/decision-making abilities.

## 2. Materials and Methods

### 2.1. Study Design and Participants

This mixed methods study included a pre- and post-survey design to evaluate undergraduate nursing students’ confidence in pediatric nursing knowledge, skills, and decision-making abilities after participation in both an instructor-led (guided) and a student-led (decision-making) simulation involving unfolding case-study scenarios. This study was conducted at a public, rural, liberal arts university in the Southeastern United States. The study was conducted according to the guidelines of the Declaration of Helsinki and was approved by the University Institutional Review Board (IRB) (protocol #11-05-202005, 8 December 2020). A signed informed consent statement was waived for this study by the IRB due to the voluntary nature of the survey. Students read a description of the study before volunteering to participate. Students completed the survey voluntarily and anonymously. No participant identifiers were collected. All demographic data were aggregated to determine the general characteristics of the participant sample.

The undergraduate bachelor of science in nursing (BSN) program is comprised of two years of nursing prerequisite courses, followed by admission to the upper division program and completion of two years of nursing-specific courses. The upper division program is typically comprised of approximately 230 nursing students. The undergraduate nursing pediatrics course is taught in the third/Senior I semester of the upper division program.

The participants in this study included a convenience sample of forty-three BSN/Senior I level students. All students were enrolled in the undergraduate nursing pediatrics course and were required to participate in two high-fidelity clinical simulations as part of the course’s clinical component. Students had previously participated in a simulation scenario with a standardized patient in their health assessment course, but none had experience with high-fidelity simulations during the nursing program before this experience. The simulations served as formative clinical assessments and did not influence course grades. Student participation in the pre- and post-simulation surveys was voluntary.

Because securing pediatric clinical sites during the COVID-19 pandemic became more challenging and some previously utilized sites were unavailable, there was an increased need to incorporate additional experiences for the students through simulation The objectives of the simulations were to improve student knowledge, self-confidence, nursing skills, communication skills, and decision-making/clinical judgement abilities associated with the care of the hospitalized pediatric patient. This was accomplished using two separate simulations with unfolding case-study scenarios. The first simulation occurred during the first clinical day of the semester and was instructor-led (guided) in nature. The adjunct clinical instructor provided teaching throughout the simulation scenario and guided students through three consecutive cycles of appropriate clinical decisions and care interventions based on patient presentation, needs/cues, and changes in patient condition. The adjunct instructor helped the students to refine clinical skills while critically thinking about how to make proper patient care decisions. The second simulation occurred at the end of the semester and required the students to lead the patient care scenario and to make critical decisions about best care interventions for the simulated patient. The objectives for each simulation (Table 1) aligned with both course and program outcomes [17,18]. Prior to each simulation, students were assigned activities to complete and resources to review to ensure their familiarity with the simulation content, which is supported by the International Nursing Association for Clinical Simulation and Learning (INACSL) standards of simulation design [17]. In addition, a pre-briefing was conducted immediately before each simulation to orient students to the high-fidelity simulator, to the simulation environment, and to discuss professional expectations and confidentiality [17]. The high-fidelity simulator that is used for the pediatrics course is a school-aged Black/African American pediatric patient. This is the only high-fidelity pediatric simulator available at the university. The patient diagnoses used for the simulation were abdominal pain (Diabetic Ketoacidosis) and Sickle Cell Anemia (Sickle Cell Crisis). These diagnoses are common in the region and are typical of cases that the students may encounter on the local hospital floor during their Pediatric clinical rotations. Sickle cell anemia is common in the local African-American population. Thus, a sense of realism is added by portraying a school-aged African American sickle cell patient using a high-fidelity simulator that matches the patient description. One course coordinator and one adjunct clinical instructor conducted all pediatric simulations, which provided for a consistent simulation experience for all students. Both simulation facilitators have the educational background and nursing experience in pediatrics to guide students towards meeting the learning objectives [19]. Students participated in the simulations within their assigned clinical groups, which consisted of seven to eight students. The pediatric simulation experiences served as a formative evaluation method to nurture personal and professional growth and assist the students toward meeting the previously stated objectives [20].

### 2.2. Pediatric Simulation Scenarios

The first simulation experience was an instructor-led (guided) unfolding case-study scenario that was conducted at the beginning of the semester and included a school-aged child (high-fidelity simulator) who presented to the ED in an apparent sickle cell crisis. The purpose of the instructor-led scenario was to provide a comprehensive teaching and learning experience for students with step-by-step instructions and rationales for patients care. Students completed pre-assignment questions on sickle cell disease pathophysiology, medications, and care interventions. The adjunct clinical instructor guided the students through each step of the scenario, from patient presentation in the emergency department (ED), to admission to the Pediatric Intensive Care Unit (PICU), and then discharge to home. The unfolding scenario included three cycles of patient assessment, care implementation, and management. The adjunct instructor guided students through the challenges presented in the scenario and demonstrated how to make good clinical decisions by utilizing both the traditional nursing process and the clinical judgement model (recognizing and analyzing cues to form hypotheses, taking action by prioritizing hypotheses and generating solutions, and evaluating outcomes based on observation and experience [7]. Students had an opportunity to practice nursing skills including patient vital signs/assessment, age-appropriate therapeutic communication, pain assessment and management techniques, obtaining orders from a provider (course coordinator), implementing orders and prioritizing care, medication dosage and intravenous (IV) fluid rate calculation and administration, starting a peripheral IV, setup and administration of oxygen, IV pump manipulation and programming, administering a blood transfusion, calling report to another nurse (course coordinator), and discussing discharge teaching points. This guided process provided a comprehensive introduction to the simulated hospital setting and allowed students to mentally and physically prepare to enter the hospital setting the following week.

The second simulation occurred on the final clinical day of the semester and involved a scenario of a school-aged child (high-fidelity simulator) presenting to the ED with complaints of abdominal pain. The student-led scenario included the same structure and flow as the instructor-led scenario. However, the student-led scenario was designed to allow students to work more independently using the knowledge and skills gained throughout the semester. The instructor is present to answer questions and to re-direct as needed, but students lead and determine patient care decisions, thus influencing the simulated patient’s outcomes. Students completed pre-simulation work to explore potential disease pathologies/diagnoses associated with this patient presentation. The design and flow of this student-led (decision-making) unfolding case-study scenario was similar to the first simulation. However, this experience was student-led with minimal instructor interaction. This scenario, like the first, included three cycles of patient assessment, care implementation, and management, with students making independent decisions about the prioritization of care. Students simultaneously used critical thinking skills and clinical skills to collect information needed to formulate a proper patient diagnosis and care management plan. The simulated patient’s condition could improve or decline depending on the decisions made during the simulation scenario. A debriefing session at the conclusion of the scenarios allows students to discuss lessons learned and the impact of the experience on their knowledge and skills [17].

### 2.3. Evaluation Methods

A survey was deployed through the online course management system and was available for anonymous participant completion before the first simulation experience and after both the instructor-led and student-led scenarios. Survey items captured demographic information and measures of student self-confidence in pediatric nursing knowledge, skills, and clinical judgment to assess the impact of the use of pediatric simulation. The survey was adapted from the Perceived Confidence in Pediatric Knowledge and Skills Questionnaire utilized in a prior study with permission [15]. The original survey was evaluated for content validity by an expert panel and demonstrated a high level of internal consistency reliability both overall (α = 0.97) and for each subscale/domain (α = 0.83–0.93). The authors did not further evaluate the adapted survey instrument for measures of validity or reliability. The 16-item survey was structured to evaluate the four steps/domains of the nursing process (assessment, planning, implementation, and evaluation), with four corresponding evaluative items under each domain. Survey items were scored using a 5-point Likert scale from 1 (completely lacking confidence) to 5 (very confident). Five open-ended questions were also administered with both of the post-simulation surveys to allow students an opportunity to further express their perceptions of the experiences.

### 2.4. Data Analysis

Data were analyzed using SPSS statistical packaging software version 27 (Armonk, NY, USA) [21]. Item scores were reported as medians and were analyzed as ordinal data [22]. The pre-simulation survey, instructor-led post-simulation survey, and student-led post-simulation survey responses were compared for statistically significant differences using Friedman’s ANOVA testing with an alpha set at 0.05. Post-hoc Wilcoxon signed-rank tests with Bonferroni correction were used to follow up the significant findings with an adjusted alpha of 0.0167 indicating significant differences in median score responses.

Qualitative analyses of the open-ended responses were conducted using Microsoft Excel and directed content analysis [23,24]. Directed content analysis allows for the examination of data with minimal interpretation, limiting the investigator’s pre-conceived perception or biases during the analyses. Two authors coded text responses using keywords, labeled high-frequency words as nodes, and determined appropriated corresponding categories for each node. The researchers who conducted the initial analyses did not participate in the simulation experiences, thus limiting investigator bias. Constant comparative techniques and re-examination of the nodes and categories allowed for the emergence of themes about the pediatric simulation experiences. A third investigator (course coordinator and simulation participant) reviewed and separately coded the text. A final team session was conducted to combine and refine findings, although very few differences in each independent analysis were evident.

## 3. Results

### 3.1. Student Characteristics

All 43 students who were enrolled in the pediatrics course completed the pre-simulation survey. Forty (93%) completed the instructor-led post-survey and 26 (60.5%) completed the student-led post-survey (Table 2). Characteristics of this cohort of students closely represent the typical demographic makeup of all students who are enrolled in the upper-division program. The students were mostly young and under 25 years of age (72%), female (88%), and had either no clinical work experience or less than three years of experience (90.7%). Race/Ethnicity of the class cohort included 35 (81.4%) White/Caucasian, 7 (16.3%) Black/African American, and 1 (2.3%) Asian.

### 3.2. Pediatric Nursing Knowledge, Skills, and Decision Making/Clinical Judgement Abilities

Friedman’s ANOVA analyses (Table 3) revealed that all 16-items demonstrated statistically significant differences between the three measured group responses (pre-survey, instructor-led simulation post-survey, and student-led simulation post-survey), with pre-post median range 2.0–5.0, *χ*^2^ range 13.9–41.7, and *p*-value range = <0.001–0.001. Furthermore, students’ self-perception ratings of knowledge, skills, and self-confidence improved over the duration of the semester and after each simulation experience for six of the 16 scored items (pre-post median range 2.0–5.0, *χ*^2^ range 14.7–32.2, *p*-value range = <0.001–0.001). Post-hoc Wilcoxon signed-rank tests revealed statistically significant differences in student ratings pre-simulation experience and post-instructor-led experience for all of the 16-scored items (Table 4) (pre-post median range = 2.0–4.0, *Z*-score range = −5.2–2.9, *p*-value range = <0.001–0.003). While median response scores increased throughout the semester and between simulation experiences for six of the survey items, differences in those group responses were not statistically significant after applying Bonferroni adjustment.

### 3.3. Qualitative Responses

Directed content analysis of the five open-ended survey questions revealed four themes: perception of experience, pediatric nursing care, assimilation of knowledge, and critical thinking (Table 5). Perception of Experience shifted between the first and second simulations. Prior to the first simulation, 70% (28/40) felt nervous about the unknown, and 15% (6/40) expressed feelings of excitement and anticipation. One student stated, “I did not know what to expect and was nervous about it”. However, prior to the second simulation, only 30% (8/26) expressed sentiments of nervousness, and the excitement and anticipation grew to 42% (11/26). One student remarked, “I was nervous for the first simulation, but going into the second simulation I felt much more prepared and excited”.

The second theme focused on Pediatric Nursing Care. Prior to this course, students had only taken care of and interacted with the adult population. Students conveyed feelings of increased confidence regarding the care of pediatric patients and enhanced communication skills after the simulation experience. One student commented, “I am extremely confident in my ability to take care of a pediatric patient after these simulations. They really help tie in all the information we have learned and how to put it to use in the real world”. Another student said, “This simulation improved my understanding and skill of pediatric nursing care as well as improved my communication skills between other healthcare providers”.

Assimilation of Knowledge and Critical Thinking were the final two themes. One student felt the simulations “really helped to put the whole picture together; from admission, to calling the doctor for orders, to assessing, intervening, and modifying the plan of care as needed based on the patient’s situation”. Another student voiced, “It helped bridge the gap between what we learn in lecture and how to apply it to the clinical setting”. Finally, students enjoyed the opportunity to critically think about the scenarios. One expressed, “It helped me develop my critical thinking skills to understand why each intervention is done”.

## 4. Discussion

Both quantitative data and open-ended survey responses revealed overwhelmingly positive student feedback about the impact of the simulation experiences on their knowledge, skills, and self-confidence in performing as a nurse in a pediatric hospital setting. While students found both experiences beneficial, they reported the most improvement in learning and comfort from the instructor-led simulation and the most improvement in critical thinking and assimilation of knowledge during the student-led simulation.

These findings align with current research on the benefits of pediatric simulation to improve student knowledge, self-confidence, and satisfaction [25]. Likewise, our findings are reflective of those in a previous qualitative study by Teles and researchers, wherein students felt more comfortable caring for children and their families and with the use of pediatric nursing equipment after participation in the pediatric simulations [26]. Gilfoyle and team found that pediatric resuscitation simulation-based educational interventions significantly improved clinical performance and teamwork [27]. During the student-led simulation debriefing, students voiced that they enjoyed working together with their group members to critically think and to collaborate on potential diagnoses and appropriate prioritization and planning of care. In addition, the simulation may increase the clinical judgment of students. This is supported by Sherrill, who encouraged the use of simulation as a way to apply the Clinical Judgment Model, which may increase preparedness for the Next Generation NCLEX [8].

Although Saied and Cardoza and Hood found that overall student self-efficacy decreased after simulation, we found no indicators of this in either our qualitative or quantitative analyses [13,25]. Conversely, students voiced that they felt that the pediatric simulation experiences were some of the most beneficial clinical learning opportunities they had experienced thus far in the program because of the opportunity to perform skills that they would not likely encounter in the hospital setting as a student nurse.

### Limitations

Although the simulation experiences have been conducted in the described format for multiple semesters at this university, this was the first formal evaluation of student perceptions of knowledge, skills, and satisfaction. Previously, informal feedback during debriefing sessions has been used to modify and improve the simulation structure and flow to maximize the student learning experience. The surveys were conducted in an anonymous manner with no academic penalty for non-participation. However, it is possible that students felt obligated to participate or to respond positively to the survey questions because of their status as students enrolled in the course. Future analyses across multiple cohorts and locations could provide further insight as to the benefit of these experiences and what elements should be added to enhance student critical thinking and assimilation of knowledge. Additionally, students should be tracked and surveyed six-twelve months post-graduation to determine the impact of the simulation experiences on preparation for practice. Some students chose not to complete the final survey, which was held during the last week of the semester. Students voiced being fatigued and time restraints as reasons for not completing the final study. One student offered the suggestion to add an actor as a family member during the scenarios to help students navigate therapeutic communication with the family.

## 5. Conclusions

Simulation experiences that are incorporated throughout the nursing curriculum can improve student knowledge, performance, and preparation for practice. Additionally, simulation provides students an opportunity to gain experience in areas where clinical placements are scarce. Pediatric simulation experiences are important to positively impact both the learning and self-confidence of undergraduate nursing students. While simulation experiences can be conducted in a variety of ways, the incorporation of high-fidelity simulators and equipment with simultaneous patient care management during a given scenario may maximize students’ clinical judgment and assimilation of knowledge. Incorporation of unfolding case-study scenarios into simulation experiences can improve students’ self-confidence in nursing knowledge and skills and can enhance decision-making/clinical judgement abilities needed for success on both the Next Generation NCLEX-RN exam and in practice as a registered nurse.

## Figures and Tables

**Table 1 healthcare-09-01584-t001:** Simulation Objectives.

Instructor-Led Simulation Objectives (Sickle-Cell Scenario)	Student-Led Simulation Objectives (Abdominal Pain Scenario)
1. Performs accurate vital signs and respiratory and cardiac focused assessment in the pediatric simulated hospital setting (application).	1. Performs accurate vital signs and conducts a complete physical assessment in the pediatric simulated hospital setting (application).
2. Discusses pathophysiology related to sickle cell and respiratory and cardiac assessment (comprehension).	2. Discusses potential disease pathologies/diagnoses related to abdominal pain andpatient presentation/assessment (comprehension).
3. Demonstrates appropriate management and care of child with Sickle Cell Crisis (application).	3. Demonstrates appropriate management and care of a child with abdominal pain and associated disease pathologies/diagnoses (application).
4. Interprets appropriate laboratory and diagnostic tests for the management of Sickle Cell Crisis and Acute Chest Syndrome (analysis).	4. Interprets appropriate laboratory and diagnostic tests for the management of abdominal pain and associated disease pathologies/diagnoses (analysis).
5. Demonstrates appropriate therapeutic com- munication with a school-aged child (application).	5. Demonstrates appropriate therapeutic communication with a school-aged child (application).
6. Determines effectiveness of care and pain management (Evaluation).	6. Determines effectiveness of care and pain management (Evaluation).
7. Recognizes complications of ineffective sickle cell treatment and long-term effects of chronic illness in children (comprehension).	7. Formulates an appropriate plan of care for the pediatric patient in the simulated hospital setting based on diagnosis (Synthesis).

**Table 2 healthcare-09-01584-t002:** Student Demographics.

Variable	Pre-Simulation Pre-Survey, (*n* = 43), *n* (%)	Simulation #1 (Instructor-led) Post-Survey #1, (*n* = 40), *n* (%)	Simulation #2 (Student-Led) Post-Survey #2, (*n* = 26), *n* (%)
Gender			
Female	38 (88.4)	35 (87.5)	23 (88.5)
Male	5 (11.6)	5 (12.5)	3 (11.5)
Race/Ethnicity			
White/ Caucasian	35 (81.4)	32 (80.0)	23 (88.5)
Black/African American	7 (16.3)	7 (17.5)	3 (11.5)
Asian	1 (2.3)	1 (2.5)	0 (0.0)
Age			
18–25 years	31 (72.1)	31 (77.5)	20 (76.9)
26–35 years	8 (18.6)	6 (15.0)	3 (11.5)
36–45 years	3 (7.0)	2 (5.0)	3 (11.5)
46–55 years	1 (2.3)	1 (2.5)	0 (0.0)
Clinical Work Experience			
None	7 (16.3)	7 (17.5)	5 (19.2)
<1 year	13 (30.2)	13 (32.5)	6 (23.1)
1–3 years	19 (44.2)	18 (45.)	13 (50.0)
4–6 years	3 (7.0)	1 (2.5)	2 (7.7)
7–10 years	1 (2.3)	1 (2.5)	0 (0.0)

**Table 3 healthcare-09-01584-t003:** Presurvey, Instructor-led Postsurvey, and Student-led Postsurvey Responses (Medians).

Item	Pre	Instructor-Led Post	Student-Led Post	*χ^2^*	*p*
In the simulated hospital setting, how confident are you in…					
Assessment					
Knowledge of correct pediatric health assessment techniques on a child	2.0	4.0	4.0	37.8	<0.001
Performing effective health assessments on a child	2	4	4	41.7	<0.001
Use of communications skills to form effective, collaborative partnerships with children and their families.	3	4	5	32.2	<0.001
Ability to accurately describe/document the assessment of a child	2.0	4.0	5.0	28.4	<0.001
Planning					
Knowledge about plans for pediatric nursing care related to the assessment of a child	2.0	4.0	4.0	36.0	<0.001
Ability to display skills in critical thinking and ethical decision-making in health promotion and health protection strategies with children and their families	3.0	4.0	4.0	27.1	<0.001
Competency in communicating a nursing plan to families of different cultures, communities, and complexities	3.0	4.0	4.0	22.1	<0.001
Ability to accurately describe appropriate nursing plans for a child	3.0	4.0	4.5	28.1	<0.001
Implementation					
Knowledge of the multifaceted roles of the nurse in promotion and protection of the health of children and their families in a simulated hospital environment (caregiver, advocate, communicator, educator).	3.0	4.0	4.5	26.9	<0.001
Ability to use technical skills in a timely and effective manner in simulation experiences (includes nursing interventions, isolation, and universal precautions).	3.5	4.0	5.0	14.7	0.001
Ability to collaborate with other health care professionals in health promotion/health protection for children and families	4.0	4.0	5.0	13.9	0.001
Ability to document follow-up changes in pediatric patient condition	3.0	4.0	5.0	22.6	<0.001
Evaluation					
Knowledge of the “next step” when a child’s condition changes either expected or unexpectedly	2.5	4.0	4.0	31.9	<0.001
Ability to adapt to changes and modify nursing plans for a child and family	3.0	4.0	4.0	18.1	<0.001
Ability to verbalize evaluation of the nursing roles of a caregiver, advocate, communicator, and educator in simulation discussions and debriefing.	3.0	4.0	4.0	14.9	0.001
Ability to accurately reassess and communicate whether outcomes of nursing care of a child and family in a simulated hospital setting are met or how they need modification if not met.	2.0	4.0	4.0	19.7	<0.001

Analyzed using Friedman’s ANOVA.

**Table 4 healthcare-09-01584-t004:** Presurvey and Instructor-led Postsurvey Responses (Medians).

Item	Pre	Instructor-Led Post	*Z*	*p*
In the simulated hospital setting, how confident are you in…				
Assessment				
Knowledge of correct pediatric health assessment techniques on a child	2.0	4.0	−5.1	<0.001
Performing effective health assessments on a child	2	4	−5.2	<0.001
Use of communications skills to form effective, collaborative partnerships with children and their families	3	4	−4.5	<0.001
Ability to accurately describe/document the assessment of a child	2.0	4.0	−4.8	<0.001
Planning				
Knowledge about plans for pediatric nursing care related to the assessment of a child	2.0	4.0	−5.1	<0.001
Ability to display skills in critical thinking and ethical decision-making in health promotion and health protection strategies with children and their families	3.0	4.0	−4.5	<0.001
Competency in communicating a nursing plan to families of different cultures, communities, and complexities	3.0	4.0	−4.5	<0.001
Ability to accurately describe appropriate nursing plans for a child	3.0	4.0	−4.9	<.001
Implementation				
Knowledge of the multifaceted roles of the nurse in promotion and protection of the health of children and their families in a simulated hospital environment (caregiver, advocate, communicator, educator).	3.0	4.0	−4.7	<0.001
Ability to use technical skills in a timely and effective manner in simulation experiences (includes nursing interventions, isolation, and universal precautions).	3.5	4.0	−3.5	<0.001
Ability to collaborate with other health care professionals in promotion/ health protection for children and families	4.0	4.0	−2.9	0.003
Ability to document follow-up changes in pediatric patient condition	3.0	4.0	−4.5	<0.001
Evaluation				
Knowledge of the “next step” when a child’s condition changes either expected or unexpectedly	2.5	4.0	−5.2	<0.001
Ability to adapt to changes and modify nursing plans for a child and family	3.0	4.0	−4.5	<0.001
Ability to verbalize evaluation of the nursing roles of a care giver, advocate, communicator, and educator in simulation discussions and debriefing.	3.0	4.0	−3.9	0.001
Ability to accurately reassess and communicate whether outcomes of nursing care of a child and family in a simulated hospital setting are met or how they need modification if not met.	2.0	4.0	−4.2	<0.001

Analyzed using Friedman’s ANOVA.

**Table 5 healthcare-09-01584-t005:** Participant Responses and Coding Scheme.

Student Response	Code/Node	Category	Theme
“I did not know what to expect and was nervous about it”.	Unsure and nervous	Nervous about unknown	Perception of Experience
“I was nervous for the first simulation but going into the second simulation I felt much more prepared and excited”.	Nervous for first simulation but excited for second simulation	Excited and eager for more autonomous experience	Perception of Experience
“I feel much more confident going into a pediatric care setting after participating in this simulation. I am more confident with going to the Peds clinical after this simulation”.	Increased confidence after simulation experiences	Increased confidence in pediatric nursing clinical skills	Pediatric Nursing Care
“I am extremely confident in my ability to take care of a pediatric patient after these simulations. They really help tie in all the information we have learned and how to put it to use in the real world”.	Increased confidence in pediatric skills and patient care	Increased confidence in pediatric nursing clinical skills	Pediatric Nursing Care
“This simulation improved my understanding and skill of pediatric nursing care as well as improved my communication skills between other healthcare providers”.	Improved communication with other providers	Opportunity to communicate and collaborate	Pediatric Nursing Care
“It really helped to put the whole picture together; from admission, to calling the doctor for orders, assessing, intervening, and modifying the plan of care as needed based on the patient’s situation”.	Putting together the care process	Planning and modifying the plan of care	Assimilation of Knowledge
“It helped bridge the gap between what we learn in lecture and how to apply it to the clinical setting”.	Combining knowledge	Translation of knowledge to practice	Assimilation of Knowledge
“I really enjoyed the chance to critically think about what could be wrong with the patient when [the patient] presented with symptoms”.	Opportunity to critically think about symptoms	Critical thinking and nursing care	Critical Thinking
“It helped me develop my critical thinking skills to understand why each intervention is done”.	Opportunity to critically think about interventions	Critical thinking and nursing care	Critical Thinking

## Data Availability

Data may be made available upon request from the corresponding author.

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
