# Peer review of "Assessing the Impact of Unfolding Case Study Scenarios during High-Fidelity Pediatric Simulation among Undergraduate Nursing Students"

_healthcare, 2021, doi:10.3390/healthcare9111584_

Round 1
Reviewer 1 Report
The study is interesting and relevant especially in the background of the COVID pandemic. Also, pediatric is a subspecialty that's extremely difficult for new learners and trainees because it requires attention to detail as you are taking care of kids age 0 to 21 years.
I know this was a voluntary process, but worry about the pressure that students must have felt to be part of the study even though their grades were not impacted. This is a huge commitment for a student to be part of the study and fill out the long questionnaire. As student, the natural tendency is to please your instructor and take part in the study. This should be discussed in limitations of the study.
Also, I didn't quite understand the rational for having student led simulation. It would have been interesting to see the results of instructor led simulation after the first and second scenario. You should explain the rationale for have different leads for scenarios.
Also, it maybe worth explaining the rationale for choosing these particular scenarios. Is sickle cell crisis more prevalent in your region?
For future consideration, the authors should discuss the long term impact of the high fidelity simulation on the student's nursing career. A survey once the student has graduated and is in clinical practice may add to the current literature.
Author Response
Reviewer 1
- The study is interesting and relevant especially in the background of the COVID pandemic. Also, pediatric is a subspecialty that's extremely difficult for new learners and trainees because it requires attention to detail as you are taking care of kids age 0 to 21 years.
Response: Thank you for your feedback.
- I know this was a voluntary process, but worry about the pressure that students must have felt to be part of the study even though their grades were not impacted. This is a huge commitment for a student to be part of the study and fill out the long questionnaire. As student, the natural tendency is to please your instructor and take part in the study. This should be discussed in limitations of the study. Added potential bias due to student enrollment in the course in limitations section. Lines 309-312.
- Also, I didn't quite understand the rational for having student led simulation. It would have been interesting to see the results of instructor led simulation after the first and second scenario. You should explain the rationale for have different leads for scenarios.
Added instructor-led explanation in lines 148-149: The purpose of the instructor-led scenario was to provide a comprehensive teaching and learning experience for students with step-by-step instructions and rationales for patients care.
Added student-led explanation in lines 172-176: The student-led scenario included the same structure and flow as the instructor-led scenario. However, the student-led scenario was designed to allow students to work more independently using the knowledge and skills gained throughout the semester. The instructor is present to answer questions and to re-direct as needed, but students lead and determine patient care decisions, thus influencing the simulated patient’s outcomes.
- Also, it maybe worth explaining the rationale for choosing these particular scenarios. Is sickle cell crisis more prevalent in your region?
Added to lines 135-140. Typical diagnoses for the region. These diagnoses are common in the region and are typical of cases that the students may encounter on the local hospital floor during their Pediatric clinical rotations. Sickle cell anemia is common in the local African-American population. Thus, a sense of realism is added by portraying a school-aged African American sickle cell patient using a high-fidelity simulator that matches the patient description
For future consideration, the authors should discuss the long term impact of the high fidelity simulation on the student's nursing career. A survey once the student has graduated and is in clinical practice may add to the current literature. Added plans to track and re-evaluate students 6-12 months post-graduation in lines 328-331.
Reviewer 2 Report
Dear Auhtors, congratulations for the study.
1.the introduction lacks of the theme of anxiety in nursing school students, which is reflected in the practice quality and Standard of Care.
2.Therefore, innovation technology and simulation is demanding from the first years.
3.In particular the biomedical sciences curricula can create in nursing students anxiety and a reduced performance during exams , and the diversification of teaching methods can help (https://www.mdpi.com/2076-3417/10/7/2357;
https://journals.sagepub.com/doi/10.1177/0898010112462067;
https://www.degruyter.com/document/doi/10.1515/ijnes-2017-0042/html)"
Author Response
Reviewer 2
1.the introduction lacks of the theme of anxiety in nursing school students, which is reflected in the practice quality and Standard of Care. Therefore, innovation technology and simulation is demanding from the first years. In particular the biomedical sciences curricula can create in nursing students anxiety and a reduced performance during exams , and the diversification of teaching methods can help (https://www.mdpi.com/2076-3417/10/7/2357;
https://journals.sagepub.com/doi/10.1177/0898010112462067;
https://www.degruyter.com/document/doi/10.1515/ijnes-2017-0042/html)"
Added verbiage in the introduction about student anxiety and included information from the suggested references. Citations and references were added appropriately.
Reviewer 3 Report
First of all, I would like to thank you for the opportunity to contribute some improvement to this manuscript.
The article proposed by the authors for publication in Healthcare magazine entitled: "Assessing the Impact of Unfolding Case Study Scenarios during High-Fidelity Pediatric Simulation among Undergraduate Nursing Students" deals with an interesting and current topic at the moment in the Universities where they are taught nursing studies. Clinical simulation allows students to learn in environments very similar to clinical environments, where they can learn technical skills, function in a hospital environment and even communicate with the patient. All this, emphasizing that they do it in a safe environment, where any mistake is learning and does not affect the patient, or her family, or other health professionals.
Below I detail the points of the article and mention if some minimum aspects are needed that could improve this manuscript:
1. The introduction includes the essentials to be able to know what clinical simulation is in nursing, also focusing on pediatrics. In the end it ends the objective of this study.
2. In the material and methods section, the methodology used is described in detail. Although, the section on ethical considerations (lines 85 onwards) would put them in a separate section (ethical considerations).
3. The results are well described. I would improve the tables by centering certain data, which is not aligned with the others.
I would also like you to explain to me what the n with a sidebar refers to in the lines (249, 250, 252, 253), those two numbers that appear.
4. In the discussion section, I would reduce the first summary paragraph, since it is not the section for this content.
5. In the references section, check the volume of the references, it must be in italics.
Thanks.
Author Response
Reviewer 3
First of all, I would like to thank you for the opportunity to contribute some improvement to this manuscript.
The article proposed by the authors for publication in Healthcare magazine entitled: "Assessing the Impact of Unfolding Case Study Scenarios during High-Fidelity Pediatric Simulation among Undergraduate Nursing Students" deals with an interesting and current topic at the moment in the Universities where they are taught nursing studies. Clinical simulation allows students to learn in environments very similar to clinical environments, where they can learn technical skills, function in a hospital environment and even communicate with the patient. All this, emphasizing that they do it in a safe environment, where any mistake is learning and does not affect the patient, or her family, or other health professionals.
Below I detail the points of the article and mention if some minimum aspects are needed that could improve this manuscript:
- The introduction includes the essentials to be able to know what clinical simulation is in nursing, also focusing on pediatrics. In the end it ends the objective of this study.
Response: Thank you for your positive feedback.
- In the material and methods section, the methodology used is described in detail. Although, the section on ethical considerations (lines 85 onwards) would put them in a separate section (ethical considerations).
Response: We created an Ethical Considerations section.
- The results are well described. I would improve the tables by centering certain data, which is not aligned with the others.- modified table formatting
I would also like you to explain to me what the n with a sidebar refers to in the lines (249, 250, 252, 253), those two numbers that appear. n=number. n was removed to limit confusion. - In the discussion section, I would reduce the first summary paragraph, since it is not the section for this content. -Deleted sentence about open-ended responses to reduce content in the first paragraph of the section
- In the references section, check the volume of the references, it must be in italics.
Response: The volume has been italicized.